# Horizontal-to-Vertical Spectral Ratio of Ambient Vibration Obtained with Hilbert–Huang Transform

**DOI:** 10.3390/s21093292

**Published:** 2021-05-10

**Authors:** Maik Neukirch, Antonio García-Jerez, Antonio Villaseñor, Francisco Luzón, Mario Ruiz, Luis Molina

**Affiliations:** 1Geosciences Barcelona, GEO3BCN-CSIC, C/Lluis Solé i Sabarís s/n, 08028 Barcelona, Spain; mruiz@geo3bcn.csic.es; 2Department of Chemistry and Physics, University of Almería, Carretera Sacramento s/n, La Cañada de San Urbano, 04120 Almería, Spain; agarcia-jerez@ual.es (A.G.-J.); fluzon@ual.es (F.L.); 3Institute of Marine Sciences, ICM-CSIC, Passeig Marítim de la Barceloneta 37-49, 08003 Barcelona, Spain; antonio.villasenor@csic.es; 4Department of Biology and Geology, University of Almería, Carretera Sacramento s/n, La Cañada de San Urbano, 04120 Almería, Spain; lmolina@ual.es

**Keywords:** HVSR, non-stationary, data processing

## Abstract

The Horizontal-to-Vertical Spectral Ratio (HVSR) of ambient vibration measurements is a common tool to explore near surface shear wave velocity (Vs) structure. HVSR is often applied for earthquake risk assessments and civil engineering projects. Ambient vibration signal originates from the combination of a multitude of natural and man-made sources. Ambient vibration sources can be any ground motion inducing phenomena, e.g., ocean waves, wind, industrial activity or road traffic, where each source does not need to be strictly stationary even during short times. Typically, the Fast Fourier Transform (FFT) is applied to obtain spectral information from the measured time series in order to estimate the HVSR, even though possible non-stationarity may bias the spectra and HVSR estimates. This problem can be alleviated by employing the Hilbert–Huang Transform (HHT) instead of FFT. Comparing 1D inversion results for FFT and HHT-based HVSR estimates from data measured at a well studied, urban, permanent station, we find that HHT-based inversion models may yield a lower data misfit $\chi^{2}$ by up to a factor of 25, a more appropriate Vs model according to available well-log lithology, and higher confidence in the achieved model.

## 1. Introduction

The Horizontal-to-Vertical Spectral Ratio (HVSR) of ground motion measurements is a common tool to characterize near surface shear wave (Vs) structure [1,2,3,4]. It is often applied for earthquake risk assessments [5,6] and civil engineering [7,8]. HVSR relies on ambient noise as time series that can be measured with common three-component (3C) seismometers or, recently, by a combination of a seismometer for the vertical motion and fiber optic cables for finely distributed array sensing for the horizontal component [8] or even from active source seismics [9].

Ambient vibration refers to persistent, broad band ground vibrations caused by a multitude of natural and/or man-made sources. It consists mostly of elastic surface waves that originate from ocean waves, winds and human activity such as machinery, industries or public transportation to name a few [2]. Typically, the Fast Fourier Transform (FFT) is applied to obtain spectral information from the measured time series but, given the complexity and randomness of sources, the measured signal need not be sufficiently stationary and may bias FFT spectra. This may be particularly problematic in urban areas with abundant anthropogenic vibration sources of very heterogeneous origins and powers. Among the alternatives to the FFT that have been considered for non-stationary data analysis are wavelet analysis [10,11] and the Hilbert–Huang Transform [12,13]. While wavelet analysis still requires one to choose basis functions, the Hilbert–Huang Transform (HHT) introduced by [14] is truly data adaptive and therefore, well suited for the analysis of general time series of unknown origin that include non-stationarity. HHT, and its fundamental engine, the Empirical Mode Decomposition (EMD), have become a widely used tool to analyze time series measurements that perform equally well for stationary as for non-stationary signals, e.g., [15,16,17,18]. Whether or not non-stationary signals should be removed from time series for HVSR processing is still debated between authors who consider it necessary to exclude spikes and transients in microtremors, e.g., [19,20,21] who suggest that non-stationary large amplitude noise windows should not be removed, because these can carry subsoil information, potentially improving the correlation between noise and earthquakes in HVSR curves. Furthermore, other authors suggest methods for the selection of time windows using statistical criteria that use agglomerative hierarchical clustering, e.g., [22,23].

EMD describes the time series based on a data-adaptive spectral basis which may vary over time, therefore each channel of a 3C signal is not necessarily described on the same spectral basis at all times if each component is treated as univariate, i.e., independently. This is not a problem for the FFT, which describes a time series by its spectral components by assuming a static spectral basis. Therefore the FFT can be univariate and, still, the basis will be uniform for a 3C signal enabling an easy calculation of the spectral ratio between different channels. Multivariate Empirical Mode Decomposition (MEMD) [24] overcomes this problem by enforcing most similar spectral bases (still data adaptive and time varying) on 3C signals and therewith allowing intrinsically a time varying spectral analysis of a non-stationary 3C signal. MEMD has been evaluated as a filter in seismology and has been reported as very effective in reducing noise for earthquake hypocenter analysis [25].

Application of the HHT for HVSR estimation has been limited to the use as a filter to remove unwanted components in the time series, which then would yield improved HVSR estimates by traditional FFT [12,13]. The reason for this is that the instantaneous spectral data of a 3C signal obtained by univariate HHT cannot be interpreted between the components due to incoherence in the spectral basis between the components. This problem can be addressed adequately by using MEMD as reported by, e.g., [25] for seismologic data and [26] for magnetotelluric data. Applying MEMD to HVSR data processing allows to utilize directly the 3C instantaneous spectral data and avoid FFT data processing altogether.

In this work, we illustrate the benefits that MEMD can bestow on HVSR data processing and data inversion in comparison to a process purely based on FFT. To this end, we first review the (M)EMD for the context of a 3C seismic signal, state our statistical framework for the computational estimation of the HVSR from spectral instantaneous parameters and lay out our strategy to compare MEMD and FFT results. Then, we demonstrate our findings exemplarily on data from two different sites, one field site and one permanent station. Using data from well studied stations, we corroborate our results with available well-log lithology. Our HHT-based inversion models for the permanent station yield a lower data misfit $\chi^{2}$ by a factor of 25, a more appropriate Vs model according to the available well-log lithology, and a higher confidence in the achieved model.

## 2. Methodology

### 2.1. Multivariate Empirical Mode Decomposition (MEMD)

EMD is used in the HHT to obtain a set of zero-mean functions for each of which a physically meaningful analytic signal is guaranteed to exist. Such a zero-mean function is referred to as intrinsic mode function (IMF, denoted by the greek letter iota, ι). Then, the analytic signal of an IMF contains the instantaneous spectral parameters (ISP), amplitude, phase and frequency that are subsequently used for spectral analyses. Let us review the basic process for EMD [14] on a time series d(t) (see Figure 1) with pseudo-code:Copy time series $d_{\mathrm{temp}}=d\left(t\right)$, initialize mode index (m=0) and choose a sifting tolerance (e.g., $10^{-6}$).Copy the time series $c=d_{\mathrm{temp}}$.Compute cubic splines through maxima and minima of *c*, these are the envelopes $c_{\max}$ and $c_{\min}$.Obtain the mean of the envelopes $c_{0}=0.5\left(c_{\max}+c_{\min}\right)$ and subtract $c_{0}$ from *c*.Continue if $\sum |c_{0}|$ is zero (up to a tolerance), otherwise return to point 3.Save current *c* (from which you repeatedly subtracted $c_{0}$) as IMF with m=m+1: $\iota_{m}=c$. Then subtract current IMF from $d_{\mathrm{temp}}$ to obtain the new, reduced time series $d_{\mathrm{temp}}=d_{\mathrm{temp}}-c$If $d_{\mathrm{temp}}$ has 3 or fewer extrema continue, otherwise return to point 2.Save $d_{\mathrm{temp}}$ as residual: $r=d_{\mathrm{temp}}$.

The EMD algorithm yields a (data dependent) number of IMFs and one residual, which will, if summed up, yield the original data exactly: (1)$$d\left(t\right)=\sum_{m=1}^{m_{\max}}\iota_{m}\left(t\right)+r\left(t\right)$$

There are a number of plausible strategies to compute the analytic signal from an IMF [28] and some may be more adequate than others, depending on the nature of the data. The choice of how to compute the analytic signal is not central to this work. Originally, the Hilbert Transform is suggested to compute the corresponding imaginary part to an IMF but [28] reports that the direct quadrature (DQ) generally provides a more robust estimate of the analytic signal. In this manuscript, we use the direct quadrature, yielding a complex-valued IMF $\tilde{\iota }$: (2)$${\tilde{\iota }}_{m}\left(t\right)=\iota_{m}\left(t\right)+i\mathrm{DQ}\left(\iota_{m}\left(t\right)\right).$$

Time series of instantaneous spectral amplitude α, phase ϕ and frequency ω can be derived from the IMF’s analytic signal as magnitude, argument and the argument’s time derivative, respectively [28]: (3)$${\tilde{\iota }}_{m}\left(t\right)=\alpha_{m}\left(t\right)e^{2i\pi \phi_{m}\left(t\right)}\;\mathrm{and}\;\omega_{m}\left(t\right)=2\pi \frac{\partial \phi_{m}\left(t\right)}{\partial t}$$

As mentioned before, the HVSR compares the spectra of horizontal and vertical ground motion measurements and, therefore requires for the spectra to be on the same spectral basis, which they are by default for the FFT. The spectral basis for ISP obtained by EMD are data dependent and it cannot be guaranteed that two different measurements, i.e., horizontal and vertical components, yield the same spectral basis. To overcome this kind of problem, Ref. [24] proposed a multivariate expansion to the EMD algorithm, referred to as MEMD. Typical 3C seismic ground motion measurements are trivariate signals. The multivariate variant EMD projects the multicomponent signal on hyperplanes of orthogonal sequences and performs the sifting process on these projections. Then the superposition of the sifted projections yields the multivariant IMFs which enjoy the same properties as IMFs obtained by the original EMD, specifically that physically meaningful ISPs can be generated. The advantage over the original EMD is that MEMD produces IMFs with similar instantaneous frequency, so that the multivariate signal corresponds to a common spectral basis at all times. For more details we refer to the work by [24].

### 2.2. Robust, Weighted Statistics for HVSR Processing on a Logarithmic Scale

#### 2.2.1. Preliminaries

Subscripts of lowercase letters denote the elements of a matrix *x*, e.g., $x_{f,w,\cdots }$, while capitalized subscripts distinguish cardinal directions, i.e., *E* and *N* for east–west and north–south, respectively. Superscripts identify elements of a list *y*, e.g., $y^{f,w,\cdots }$ which may be differently sized matrices themselves. The sum, mean and median of a matrix *x*, with elements $x_{f,w,\cdots }$, over its dimension *w* are denoted by $\sum_{w}x_{f,w,\cdots }$, ${\mathrm{mean}}_{w}\left(x_{f,w,\cdots }\right)$ and ${\mathrm{median}}_{w}\left(x_{f,w,\cdots }\right)$, respectively, while the mean average deviation (MAD) to the mean, square root deviation (SRD) to the median and normalization are, respectively, defined by: (4)$${\mathrm{mad}}_{w}\left(x_{f,w,\cdots }\right)={\mathrm{mean}}_{w}\left(\right|x_{f,w,\cdots }-{\mathrm{mean}}_{w}\left(x_{f,w,\cdots }\right)\left|\right),$$
(5)$${\mathrm{srd}}_{w}\left(x_{f,w,\cdots }\right)=\sqrt{\left|x_{f,w,\cdots }-{\mathrm{median}}_{w}\left(x_{f,w,\cdots }\right)\right|},$$
(6)$${\mathrm{normalise}}_{w}\left(x_{f,w,\cdots }\right)=\frac{x_{f,w,\cdots }}{\sum_{w}x_{f,w,\cdots }}.$$

It is worth noting here that MAD and SRD are robust statistical instruments implemented to weight data and safeguard from strong outliers. We use MAD as robust estimate of the standard deviation and its square for the variance. Later we will assign data weights based on inverse variances, which are in fact estimated by MAD. The SRD describes how close a datum point is to the expected value of the entire data set and we transform this information to a data weight similar to a typical weight function based on residuals (deviations) in robust regression, where outliers receive a smaller weight that is calculated from a function of the inverse deviation (e.g., Huber weights).

Lastly, given weights $\rho_{f,w}$ for each datum, the weighted covariance matrix [29] for data matrix $x_{f,w}$ at time window *w* and frequency bins $f_{1}$ and $f_{2}$ is defined by: (7)$$C_{f_{1},f_{2}}\left(\rho_{f,w},x_{f,w}\right)=\frac{\sum_{w}\left(x_{f_{1},w}-\sum_{w}\left(\rho_{f_{1},w}x_{f_{1},w}\right)\right)\left(\rho_{f_{2},w}\left(x_{f_{2},w}-\sum_{w}\left(\rho_{f_{2},w}x_{f_{2},w}\right)\right)\right)}{1-\sum_{w}\rho_{f_{2},w}^{2}}.$$

#### 2.2.2. HVSR Processing Scheme

Given a 3C time series of ground motion at a location, we refer to e(t), n(t) and z(t) to the time domain measurements in east–west, north–south and vertical direction, respectively. Before the processing, we divide the time series into *W* equal windows (e.g., 15 min intervals) and prepare *F* logarithmically equally spaced frequency bins over the period range of interest.

We compute the ISPs for the trivariate signal $\left[e^{w}\left(t\right),n^{w}\left(t\right),z^{w}\left(t\right)\right]$ for each window *w* by means of MEMD and assign the instantaneous spectral amplitudes to a corresponding frequency bin *f* based on the common instantaneous frequency. We take into account that the assigned ISPs in each bin are independent and identically distributed by selecting only one data point between zero crossings, following instructions by [26].

Let us denote the binned spectral amplitudes for all windows by $E^{f,w}$, $N^{f,w}$ and $Z^{f,w}$. The exact number of samples, s(f,w), within each $E^{f,w}$, $N^{f,w}$ and $Z^{f,w}$ varies depending on the signal found in each window but, generally, lower frequencies will contain less samples while different windows at the same frequency will contain a similar amount of samples. Then, lists of directional (natural) logarithmic HVSR of all samples at each window *w* and for each frequency bin *f* are given by: (8)$$\Lambda^{f,w,E}=\ln\left(E^{f,w}\right)-\ln\left(Z^{f,w}\right)\;\mathrm{and}\;\Lambda^{f,w,N}=\ln\left(N^{f,w}\right)-\ln\left(Z^{f,w}\right).$$

Means and MADs are computed over all samples as representative values for each window and frequency:(9)$$\begin{array}{c} {\bar{\Lambda }}_{f,w,E}={\mathrm{mean}}_{s(f,w)}\left(\Lambda_{s(f,w)}^{f,w,E}\right)\;\mathrm{and}\;{\bar{\Lambda }}_{f,w,N}={\mathrm{mean}}_{s(f,w)}\left(\Lambda_{s(f,w)}^{f,w,N}\right) \end{array}$$(10)$$\begin{array}{c} \Delta_{f,w,E}={\mathrm{mad}}_{s(f,w)}\left(\Lambda_{s(f,w)}^{f,w,E}\right)\;\mathrm{and}\;\Delta_{f,w,N}={\mathrm{mad}}_{s(f,w)}\left(\Lambda_{s(f,w)}^{f,w,E}\right) \end{array}$$

Considering each element of $\bar{\Lambda }$ as a datum for the logarithmic, root squared average HVSR data yield: (11)$${\bar{\lambda }}_{f,w}=0.5\ln\left(\exp\left(2{\bar{\Lambda }}_{f,w,E}\right)+\exp\left(2{\bar{\Lambda }}_{f,w,N}\right)\right).$$

Additionally to the datum’s confidence estimate Δ from the MAD over all samples within each window, we compute for each datum the SRD to the data median over all windows with (Equation 5): (12)$$\delta_{f,w,E}={\mathrm{srd}}_{w}\left({\bar{\Lambda }}_{f,w,E}\right)\;\mathrm{and}\;\delta_{f,w,N}={\mathrm{srd}}_{w}\left({\bar{\Lambda }}_{f,w,N}\right).$$

While the MAD robustly and independently describes each datum’s confidence based on each window’s spectral amplitudes (datum precision), the SRD provides a robust confidence estimate for each datum’s reliability across all windows (datum accuracy). We combine both to an overall confidence estimate
(13)$$c_{f,w}={\left(\delta_{f,w,E}\Delta_{f,w,E}^{2}\delta_{f,w,N}\Delta_{f,w,N}^{2}\right)}^{-\frac{1}{2}}$$
and assign weights ρ to each datum derived from the normalized confidence estimates with (Equation 6): (14)$$\rho_{f,w}={\mathrm{normalise}}_{w}\left(c_{f,w}\right).$$

Then, the weighted, overall, logarithmic HVSR curve is given by: (15)$$\lambda_{f}=\sum_{w}\left(\rho_{f,w}{\bar{\lambda }}_{f,w}\right),$$
the weighted covariance matrix $C\left(\rho_{f,w},{\bar{\lambda }}_{f,w}\right)$ is obtained by (Equation 7) and the weighted standard deviation for the overall logarithmic HVSR curve at each frequency is given by: (16)$$\sigma_{f}=\sqrt{C_{f,f}}.$$

Lastly, if needed, linear HVSR estimates and their lower/upper confidence bounds are, respectively,
(17)$${\mathrm{HV}}_{f}^{0}=e^{\lambda_{f}},\;{\mathrm{HV}}_{f}^{-}=e^{\lambda_{f}-\sigma_{f}}\;\mathrm{and}\;{\mathrm{HV}}_{f}^{+}=e^{\lambda_{f}+\sigma_{f}}.$$

### 2.3. Comparison between FFT- and MEMD-Based HVSR Results

Constructing realistic, complex synthetic ambient noise data is a daunting task which would require to solve a simulation with many variables. The solution for such a complex physical problem would require to set simplifying assumptions for which, in turn, any conclusions would be of limited value, as the question would remain as whether or not assumptions made for the modeling would be representative and general enough for real data. While we could easily invent a non-stationary noise source that is able to disrupt the FFT but not MEMD, e.g., [30], such noise would be very specific and would not necessarily exist in the real world. Therefore, we prefer to demonstrate the performance of MEMD on real data.

However, it is difficult to compare two processing algorithms with real data, because the true result is generally not known. In order to avoid both, constructing unrealistic synthetic data and comparing different results of real data not knowing which one is true, we opt to use real data and go one step further by inverting the processing results. The inverted $V_{S}$ models can be more easily compared. While this strategy adds more unknowns to the comparison, it allows us to make use of existing complementary data such as well-logs and Vs models from other sources with which we can corroborate the inverted models obtained from the processing results of each processing method. Specifically, in the following examples, we will assign shear wave velocity ranges taken from existing studies to concrete layers identified from well-log lithology to generate a large number of reasonable starting models. Using the starting models, the inversion results for both algorithms’ HVSR data are compared based on the achieved $\chi^{2}$, model confidence and the models’ ability to represent the known lithology qualitatively.

## 3. Examples

### 3.1. Tests at the Station ICJA at Geoscience Barcelona, Spain

#### 3.1.1. Introduction and A-Priori Information

ICJA is a permanently installed Trillium 120 broad band seismometer [31]. It is installed in the basement of the Geoscience Barcelona (GEO3BCN-CSIC) building situated in the university district in Barcelona, Spain, a dense and busy urban environment [32,33]. Reports on the site’s Vs structure [34] and a well-log with a lithologic column are available [35]. Table 1 summarizes the well-log lithology into 11 layers and displays the starting model parameter ranges for the inversions that we will describe later. The starting model parameter range has been determined based on expected values for the given lithologic units, and only the sedimentary units, i.e., silty sand and clayey sand, have been given unusually large values because [34] report Vs velocities of around $1000\;{\mathrm{ms}}^{-1}$ at these depths. During the inversions, Vs values are allowed to range from $50\;{\mathrm{ms}}^{-1}$ to $4000\;{\mathrm{ms}}^{-1}$ for all layers.

#### 3.1.2. Data Preparation and Processing

Although the station ICJA continuously collects data at 250Hz, for the purpose of testing and comparing our algorithm, we selected the 3C measurements of a single day. The chosen day is Easter Sunday 2017, a presumingly quiet day in the middle of a revered holiday season by students, staff and scientists alike. The whole day’s data have been divided into 96 intervals of 15 min.

Geopsy [36] has been used to compute the FFT-based H/V. This is a widely used toolset for this task (among many other capacities) originated from the [37] project and continuously maintained ever since. Geopsy’s HVSR tool follows the traditional approach of using windowed FFT for spectra generation and, therefore, it relies on the assumption that the measured time series are sufficiently stationary during these windows.

At 43 frequencies in the range from 0.5 to 20Hz, we obtain windowed HVSR estimates for both methods which are used to determine a weighted average result for the day, including data confidence in the form of the covariance matrix as described in the previous section. We use Geopsy’s default settings, except for the 30 s non-overlapping Tukey windows employed to obtain estimates for each 15 min interval and the computation of the total (not averaged) horizontal power in accordance with our results and the input of our inversion routine. Similarly, MEMD estimates were computed for each 15 min interval’s ISPs. It needs mentioning that our approach with MEMD took 3 h on a standard laptop (2.2 GHz 6-Core) to provide estimates, while Geopsy computation time was tens of seconds (for the entire day of data without segmenting into separate 15 min intervals which cannot be automated currently). While this ratio may improve if MEMD software were to be optimized, it would likely remain a significant discrepancy in computation time between the methods. Processing results for both methods are illustrated in Figure 2.

Results are somewhat different, with the FFT-based method estimating more details (even though smoothing according to [38] applies) than the MEMD results (which are not smoothed). Confidence estimates appear similar (judging from standard deviation) but estimates are not overlapping, suggesting considerable disagreement between the methods’ results with regard to the shape of the HVSR curve. Nevertheless, both methods agree very well with the peak frequency at around 2.7Hz, albeit the FFT results in a larger peak amplitude.

#### 3.1.3. Inversion and Comparison

We invert our HVSR processing results with the computer code by [39] following the Diffuse Field Assumption [40,41]. The used code has been slightly modified to invert for the logarithmic HVSR and to use the corresponding covariance matrix. We only use every second data point, a total of 22 data points per method, in order to aid the inversion with faster convergence. A total of 350 inversions have been run for each, the FFT- and MEMD-based, HVSR estimates with 350 random starting models bounded by the initial parameter ranges $v_{i}$ given in Table 1, where each starting model was used once for both methods. During the inversion, model parameters had to remain within bounds between 50 and $4000\;{\mathrm{ms}}^{-1}$. The inversions evaluated more than 150,000 models for each algorithm and all models contribute to the final statistics.

We define a weight $p_{r}$ to each model based on its achieved $\chi_{r}^{2}$ misfit for an algorithm’s HVSR estimate during the inversion run *r*. First, we transform $\chi^{2}$ for each method: (18)$${\tilde{\chi }}_{r}^{2}=\frac{\chi_{r}^{2}}{\min(\chi^{2})}-1,$$
where ${\tilde{\chi }}^{2}=0$ is defined as the best achieved (and therefore the practically most likely) model for the respective method. ${\tilde{\chi }}^{2}$ is used to compare the models obtain from one method and to define the weights for that method. The misfit measure $\chi^{2}$ is used to compare the performance between the two methods. Then, model weights are defined for each method by: (19)$$p_{r}={\mathrm{normalise}}_{r}\left(\exp\left(-0.5{\tilde{\chi }}_{r}^{2}\right)\right).$$

The weighted distribution of $\chi^{2}$ for each algorithm is illustrated in Figure 3.

It can be appreciated that MEMD generally achieved a much lower (factor 25) data misfit during all inversions. The HVSR forward computations, the best Vs models, the weighted (for each layer *l*) average shear wave velocity ${\bar{v}}_{l}$: (20)$${\bar{v}}_{l}=\sum_{r}v_{l,r}p_{r},$$
and the weighted standard deviation ${\bar{\sigma }}_{l}$: (21)$${\bar{\sigma }}_{l}=\sqrt{\sum_{r}\left({\left|v_{l,r}-{\bar{v}}_{l}\right|}^{2}p_{r}\right)}$$
are shown for FFT in Figure 4 and for MEMD they are displayed in Figure 5.

Both final model estimates are qualitatively similar in the sense that the three high velocity layers are required to fit the data. MEMD data inversions yield higher confidence (smaller variance estimates) suggesting that model variation is well constrained to allow the good fit (cf. data panel in Figure 5). Larger confidence intervals for FFT-based data inversions suggest that a larger range of models fit the data equally poor (cf. data panel Figure 4) and that the inversion is unable to find any model that is more appropriate for the given data. It appears to be more difficult for the inversion algorithm to find a good model to fit the FFT result presumably due to the higher detail in the curve and the particularly high estimates around the peak frequency. Given that the final model ranges agree between FFT, MEMD and our expectations from the lithologic column of the well-log, we judge that the MEMD algorithm provides the better HVSR estimates due to its better data fit and higher confidence estimates in the recovered Vs model compared to the FFT result. A reason for the better performance of the MEMD algorithm could be that, in the present urban environment, the FFT estimates are biased by the complex wavefield which may not comply sufficiently with the required stationarity assumption.

### 3.2. Tests at the Station EJDN in a Rural Area of El Ejido (Almería, Spain)

#### 3.2.1. Introduction and A-Priori Information

The site EJDN belongs to a temporal seismic array installed from 2006 to 2019 in a Mediterranean coastal plain of Spain. It was equipped with a 120 s Güralp 3ESPDC broad band seismometer. The station was installed at a rural area in the municipality of El Ejido [5]. The shallow ground structure of this area mainly consists of a series of Pliocene–Miocene sediments and soft sedimentary rocks, often overlaid by Quaternary conglomerates or Pliocene calcarenites, and below a clear impedance contrast is related to stiffer layers of Tortonian calcarenites and Triassic limestone and dolomites. Lithological data at EJDN obtained from a borehole are available from the database of wells managed by the Spanish Geological Survey (http://info.igme.es/BDAguas/ (accessed on 1 February 2021)) under the site code 2244-1-0017. Table 2 summarizes the well-log lithology into 7 layers and displays the starting model parameter ranges for the inversions that we will describe later. The starting model parameter range has been determined based on expected values for the given lithologic units and the obtained HVSR curves. During the inversions, Vs values are allowed to range from $200\;{\mathrm{ms}}^{-1}$ to $3500\;{\mathrm{ms}}^{-1}$ for all layers and each layer’s thickness is bound by layer bottom depth.

#### 3.2.2. Data Preparation and Processing

Data at the EJDN station were collected at 100Hz for an extended period of time. However, for the purpose of testing and comparing our algorithm, we selected the 3C measurements of a single day. The chosen day is Christmas day (25 December), 2016, because this is a presumably quiet Sunday with complete data for the entire 24 h. The whole day’s data have been divided into 96 intervals of 15 min and each interval has been processed by MEMD and FFT, as with the previous example.

At 55 frequencies in the range from 0.3 to 30Hz, we obtain windowed HVSR estimates for both methods which are used to determine a weighted average result for the day, including data confidence in the form of the covariance matrix as described in the previous section. Processing results for both methods are illustrated in Figure 6.

The obtained HVSR curves for the EJDN data differ between the algorithms. Confidence estimates appear generally similar (judging from standard deviation) but FFT estimates a significantly higher confidence for the peak at 8Hz. Interestingly, the azimuthal variability in amplitude of the 8Hz peak, evaluated by the FFT method is high (not shown here), suggesting non-ideal wavefield conditions in this band or non-1D behavior. Estimates between the methods rarely overlap, suggesting considerable disagreement between the methods’ results with regard to the shape of the HVSR curve. Nevertheless, both methods agree very well with the multiple peak (and trough) frequencies at around 0.6Hz, 0.9Hz, 8Hz, and 20Hz, albeit the FFT estimates generally larger peak amplitudes.

#### 3.2.3. Inversion and Comparison

For the EJDN data inversion, we used every third datum point for a total of 19 datum points. The 80 inversions have been run for FFT- and MEMD-based HVSR estimates with 80 random starting models bounded by the initial parameter ranges $v_{i}$ given in Table 2. More than 350,000 models were evaluated for each algorithm and all models contribute to the final statistics. The weighted distribution of $\chi^{2}$ is illustrated in Figure 7.

It can be appreciated that MEMD generally achieved a lower (approximately factor 2 on average) data misfit during most inversions. The HVSR forward computations, the best Vs models, the weighted (for each layer *l*) average shear wave velocity ${\bar{v}}_{l}$ and the weighted standard deviation ${\bar{\sigma }}_{l}$ are shown in Figure 8, and in Table 3 and Table 4.

Both final model estimates are qualitatively similar in the sense that the strong impedance contrasts at 3 to 4m and at the 264m interface are required to fit the data. Model confidence in both methods’ models and associated data fit are very similar. MEMD data inversions yield generally lower Vs values, probably due to the lower peak amplitudes. A notable difference between the two methods’ results are the very near surface layers from 5m to 50m. MEMD data supports high Vs (≈$1000\;{\mathrm{ms}}^{-1}$) for a thicker layer (though it is not required), while FFT data requires this potential high velocity layer (≈$1200\;{\mathrm{ms}}^{-1}$) to create a third large velocity contrast at 30m. Unfortunately, the well-log lithology record lacks the required level of detail at these depths. More data from other sources would be necessary in order to determine which processing method performed better for this site. Nevertheless, we note that the overall data misfit was better for the MEMD processing results.

## 4. Discussion and Conclusions

The Horizontal-to-Vertical Spectral Ratio (HVSR) of three component (3C) ambient vibration measurements is a common tool to explore near surface shear wave velocity (Vs) structure. HVSR is often applied for earthquake risk assessments and civil engineering projects. Recorded ambient vibration signal originates from the combination of a multitude of natural and man-made sources, which do not need to be strictly stationary even during short times.

Several aspects of ambient noise such as its azimuthal isotropy, e.g., [42], the energy partition between different vibration modes, e.g., [43] and the stationarity, e.g., [44,45] have been investigated in previous work. Even though clear daily and weekly patterns are found in seismic noise in populated areas, e.g., [46], it often appears quasi-stationary during periods of a few hours in time scales of $10^{3}$ to $10^{4}$ s according to [42] for natural illumination. Use of longer records, stationarity test and algorithms suitable for non-stationary signals such as our HHT-based scheme are recommended for applications in urban areas.

Typically, the Fast Fourier Transform (FFT) is applied to obtain spectral information from the measured time series in order to estimate the HVSR but non-stationarity may bias the FFT spectra and HVSR estimates. We described a strategy to process ambient vibration measurements applying the Hilbert–Huang Transform (HHT) instead of FFT. The application of HHT is made possible by realizing that for 3C measurements a multivariate approach is required to ensure that all three components, i.e., the three instantaneous amplitudes, correspond to one spectral basis, i.e., the same instantaneous frequency. We lay out a robust statistic framework to obtain mean HVSR estimates and the corresponding covariance matrix directly from the instantaneous parameters through HHT without any need for FFT. The procedure is illustrated on measurements from a permanent station situated in a dense urban environment and results are compared to traditional FFT estimates. FFT and HHT results are compared by their estimates’ performance in a Vs model inversion, where the Vs models are corroborated by available well-log lithology and a vertical seismic profile. Our HHT-based inversion models yield a lower data misfit $\chi^{2}$ by a factor of up to 25, a more appropriate Vs model and a higher confidence in the achieved model in comparison to FFT-based results. While the conclusions from an experiment at only one single site may not allow to claim that the presented strategy is an improvement for general data, it does illustrate that there exist use cases for which the longer data processing time associated with our strategy may be warranted. While we present the same analysis for a second example, our available well-log lithology proved insufficiently detailed to provide conclusive arguments in favor of (or against) our strategy.

In a future work, it would be interesting to investigate if the differences between FFT- and HHT-based methods increase for “worse” illumination conditions such as anisotropic wave fields, near sources, etc., and if these methods converge to the same curve in an ideal “stationary” synthetic dataset.

## Figures and Tables

**Figure 1 sensors-21-03292-f001:**
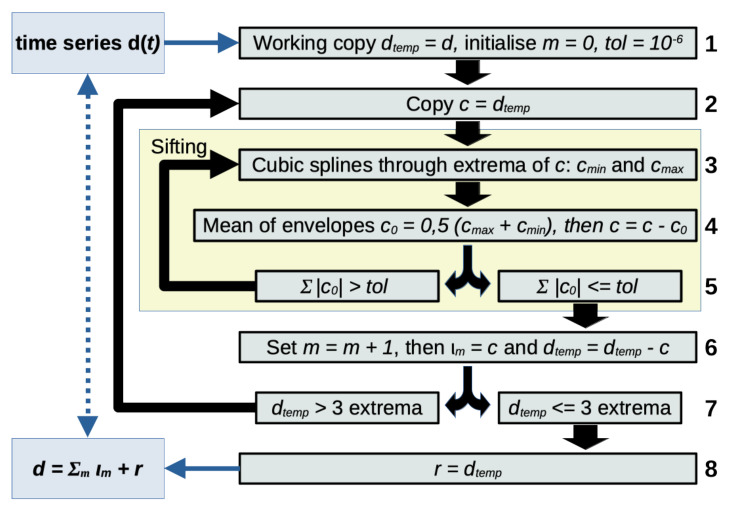
Basic Empirical Mode Decomposition. Note that the sifting stop criteria (point 5) above is given in its original form and various alternatives have been discussed in the literature [27]. However, the exact formulation of the stopping criteria for the sifting process (points 3 to 5) is not central to our work as the EMD algorithm performs with any chosen criteria.

**Figure 2 sensors-21-03292-f002:**
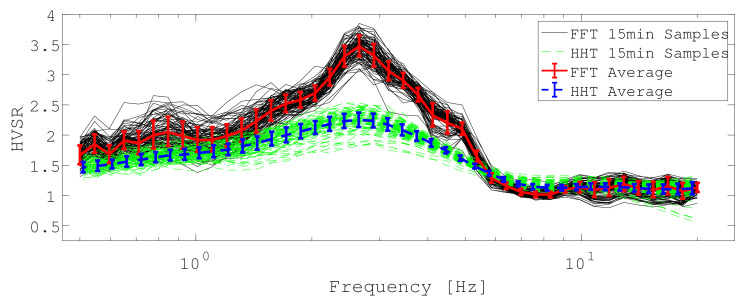
ICJA results for FFT- and MEMD-based processing.

**Figure 3 sensors-21-03292-f003:**
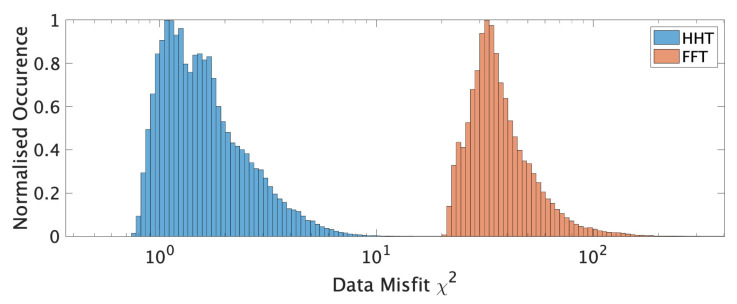
Weighted distribution of tested ICJA inversion models’ $\chi^{2}$ for FFT and MEMD curves.

**Figure 4 sensors-21-03292-f004:**
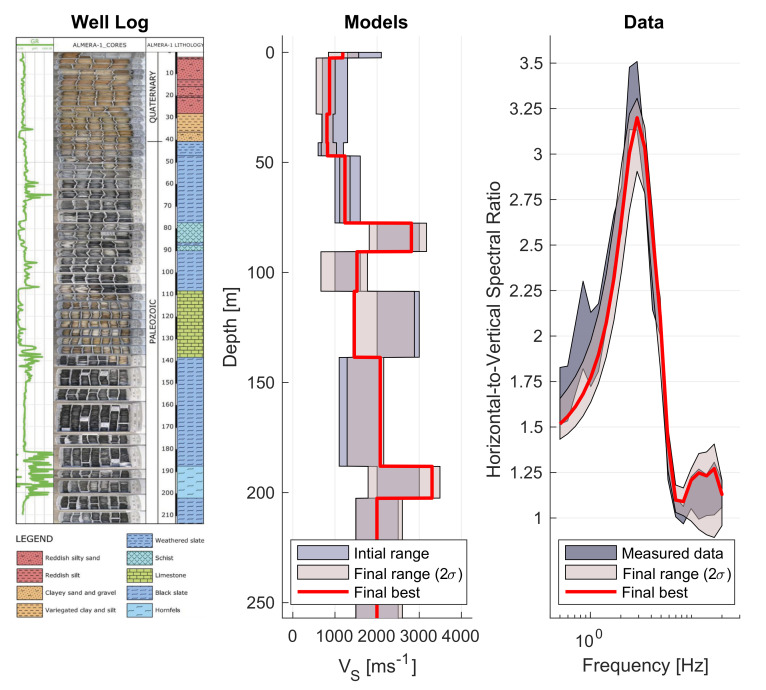
Well-log, models and data for ICJA station obtained with FFT. Well-log column taken from [35].

**Figure 5 sensors-21-03292-f005:**
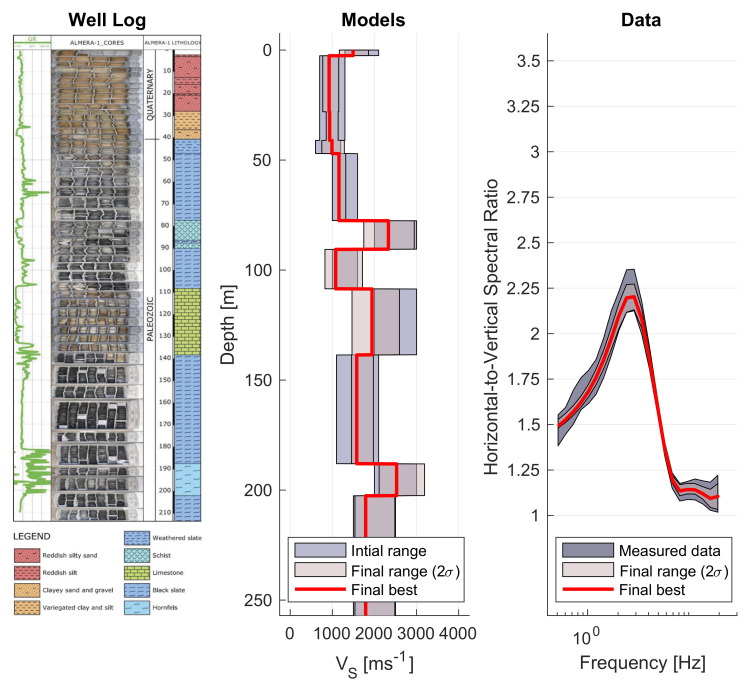
Well-log, models and data for ICJA station obtained with MEMD. Well-log column taken from [35].

**Figure 6 sensors-21-03292-f006:**
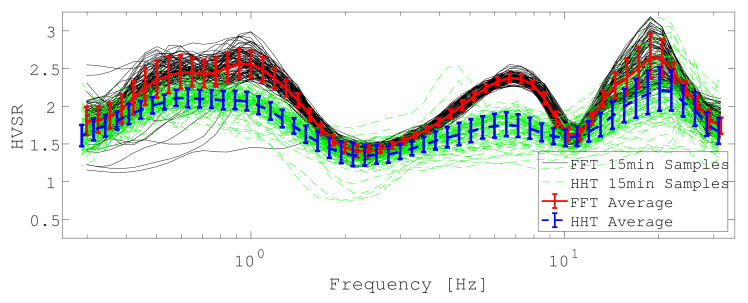
EJDN results for FFT- and MEMD-based processing.

**Figure 7 sensors-21-03292-f007:**
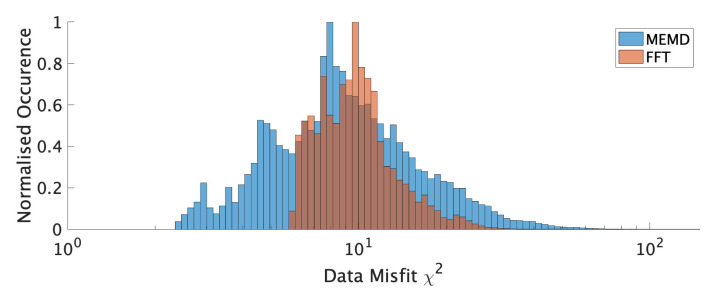
Weighted distribution of tested EJDN inversion models’ $\chi^{2}$ for FFT and MEMD curves.

**Figure 8 sensors-21-03292-f008:**
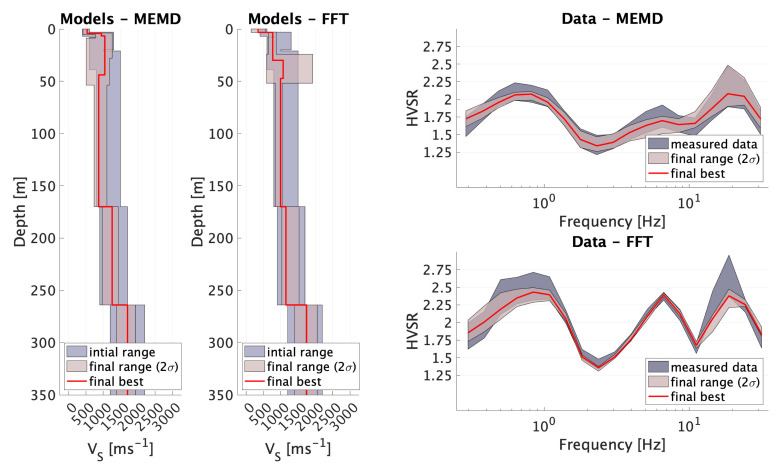
Model results and data fit for the inversion of EJDN data processed by MEMD and FFT.

**Table 1 sensors-21-03292-t001:** Well-log lithology and HVSR inversion initial and final Vs models from the ICJA station are summarized. 350 initial models were generated randomly in the range $v_{i}\pm \Delta v_{i}$. During the inversion, model parameters had to remain within the given bounds. Best model, $v_{\mathrm{best}}$, and mean model with standard deviation, $\bar{v}\pm \sigma_{v}$, resulted from the inversion of data processed by the two different algorithms, FFT and MEMD.

	Top	Starting Model Vs [m/s]		FFT Vs [m/s]		MEMD Vs [m/s]	
Lithology	[m]	$v_{i}$ ± $\Delta v_{i}$	Bounds	$v_{\mathrm{best}}$	$\bar{v}$ ± $\sigma_{v}$	$v_{\mathrm{best}}$	$\bar{v}$ ± $\sigma_{v}$
Foundation	0.0	1700 ± 400	50 to 4000	1440	1200 ± 180	1490	1510 ± 170
Silt, Sand	2.5	1000 ± 300	50 to 4000	1000	780 ± 110	920	960 ± 90
Clay, Sand	28.0	1000 ± 300	50 to 4000	920	820 ± 70	940	1000 ± 70
Weath. Sl.	41.0	900 ± 300	50 to 4000	810	850 ± 90	990	1020 ± 130
Slate.	47.0	1300 ± 300	50 to 4000	1260	1240 ±60	1160	1220 ± 50
Schist	77.5	2500 ± 500	50 to 4000	2790	2490 ± 340	2330	2350 ± 300
Slate	90.5	1300 ± 300	50 to 4000	1750	1220 ± 280	1080	1270 ± 220
Limestone	108.5	2500 ± 500	50 to 4000	2500	2160 ± 360	1940	2030 ± 280
Slate	138.5	1600 ± 500	50 to 4000	1750	1720 ± 220	1580	1720 ± 130
Hornfels	188.0	2500 ± 500	50 to 4000	3430	2640 ± 430	2530	2650 ± 270
Slate	202.5	2000 ± 500	50 to 4000	2690	2050 ± 280	1790	2010 ± 240

**Table 2 sensors-21-03292-t002:** Well-log lithology and HVSR inversion initial Vs models from the EJDN station are summarized. Initial models were generated randomly in the Vs range $v_{i}\pm \Delta v_{i}$ and bottom depth range $d_{i}\pm \Delta d_{i}$. During the inversion, model parameters had to remain within the given bounds.

	Depth [m]		Vs [m/s]	
Lithology	$d_{i}$ ± $\Delta d_{i}$	Bounds	$v_{i}$ ± $\Delta v_{i}$	Bounds
Conglomerate, Sand, Silt and Clay	5 ± 2	0 to 10	500 ± 100	200 to 3500
	14 ± 6	0 to 30	1050 ± 250	200 to 3500
Sand and Gravel	30 ± 9	0 to 150	800 ± 200	200 to 3500
Sand and Marl	170 ± 0	fixed at 170	1150 ± 350	200 to 3500
Calcarenite	264 ± 0	fixed at 264	1300 ± 400	200 to 3500
Limestone & Dolomite	950 ± 150	700 to 1200	1700 ± 500	200 to 3500
Basement	NA		2200 ± 500	200 to 3500

**Table 3 sensors-21-03292-t003:** Well-log lithology and HVSR inversion final Vs models from the EJDN station are summarized for data obtained by MEMD. The best model (depth, $d_{\mathrm{best}}$, and Vs, $v_{\mathrm{best}}$), and the mean model with standard deviation (depth, $\bar{d}\pm \sigma_{d}$, and Vs, $\bar{v}\pm \sigma_{v}$) are displayed.

Lithology	$d_{\mathrm{best}}$ [m]	$v_{\mathrm{best}}$ [m/s]	$\bar{d}$ ± $\sigma_{d}$ [m]	$\bar{v}$ ± $\sigma_{v}$ [m/s]
Conglomerate, Sand, Silt and Clay	4.2	536	4.6 ± 0.8	510 ± 48
	6.7	946	18.3 ± 8.2	1017 ± 121
Sand and Gravel	43.9	1046	40.5 ± 10.0	847 ± 169
Sand and Marl	170.0	869	170.0 ± 0.0	922 ± 93
Calcarenite	264.0	1258	264.0 ± 0.0	1204 ± 117
Limestone & Dolomite	709.8	1703	802.1 ± 149.5	1655 ± 138
Basement	NA	2061	NA	1839 ± 259

**Table 4 sensors-21-03292-t004:** Well-log lithology and HVSR inversion final Vs models from the EJDN station are summarized for data obtained by FFT. The best model (depth, $d_{\mathrm{best}}$, and Vs, $v_{\mathrm{best}}$), and the mean model with standard deviation (depth, $\bar{d}\pm \sigma_{d}$, and Vs, $\bar{v}\pm \sigma_{v}$) are displayed.

Lithology	$d_{\mathrm{best}}$ [m]	$v_{\mathrm{best}}$ [m/s]	$\bar{d}$ ± $\sigma_{d}$ [m]	$\bar{v}$ ± $\sigma_{v}$ [m/s]
Conglomerate, Sand, Silt and Clay	3.3	344	3.1 ± 0.2	348 ± 101
	30.0	767	27.3 ± 3.1	755 ± 56
Sand and Gravel	47.3	1069	44.4 ± 7.4	1254 ± 334
Sand and Marl	170.0	992	170.0 ± 0.0	940 ± 50
Calcarenite	264.0	1148	264.0 ± 0.0	1279 ± 124
Limestone & Dolomite	720.1	1744	793.2 ± 112.9	1724 ± 164
Basement	NA	1994	NA	1893 ± 266

## Data Availability

Data used in this work can be obtained by contacting labsis@geo3bcn.csic.es (for ICJA) or agarcia-jerez@ual.es (for EJDN).

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
