# Peer review of "Horizontal-to-Vertical Spectral Ratio of Ambient Vibration Obtained with Hilbert–Huang Transform"

_sensors, 2021, doi:10.3390/s21093292_

Round 1
Reviewer 1 Report
The authors develop a new method of processing microtremor data with the HVSRN method that uses the Hilbert Huang Transform, instead of the Fast Fourier Transform, to overcome problems of non-stationarity of the acquired signal. The applied methodology is described in a very rigorous way from a mathematical point of view and the approach used by the authors appears very convincing, both in terms of setting the method and in terms of the statistical analysis of the results. The choice not to use simulations, but only experimental data well constrained by geological knowledge, appears justified by the complexity of the physical problem which involves great difficulties for simulations that take into account too many physical variables.
In my opinion, therefore, the manuscript deserves to be accepted for publication on Sensors, after a minor revision that takes into account the points described below:
1) The introduction should also mention the debate among authors who consider it necessary to exclude spikes and transients in microtremors (e.g. Horike et al., 2001) and others (e.g. Mucciarelli and Gallipoli, 2003; Parolai et al., 2009) who suggest that nonstationary large amplitude noise windows should not be removed, because these can carry subsoil information, improving the correlation between noise and earthquakes in HVSRN curves. Furthermore, other authors suggest methods for the selection of time windows using statistical criteria that use agglomerative hierarchical clustering (e.g. Rodriguez and Midorikawa, 2002; D'Alessandro et al, 2016).
2) The authors should better clarify the reason for the excessive calculation time difference for the estimation of the HVSRN curve between the use of HHT compared to FFT (software not optimized or what?). In fact, this could greatly affect the preference of FFT over HHT, despite the undoubted advantages in terms of misfit obtained.
3) Even if, as mentioned above, the choice of testing the method with only experimental data appears quite justified, however the results of a single test site would not guarantee a robust evaluation of the effectiveness of the method. So I would suggest presenting at least one other test site, preferably in very different environmental conditions from the one described in the paper.
References of point 1:
D’Alessandro, A., D. Luzio, R. Martorana, and Capizzi, P. (2016). Selection of time windows in the Horizontal to Vertical Noise Spectral Ratio by means of cluster analysis. Bulletin of the Seismological Society of America, 106 (2), 560-574. doi: 10.1785/0120150017.
Horike, M., B. Zhao, and H. Kawase (2001). Comparison of site response characteristics inferred from microtremors and earthquake shear waves, Bull. Seismol. Soc. Am. 81, 1526–1536.
Mucciarelli, M., M. R. Gallipoli, and M. Arcieri (2003). The stability of the horizontal-to-vertical spectral ratio of triggered noise and earthquake recordings, Bull. Seismol. Soc. Am. 93, no. 3, 1407–1412.
Parolai, S., M. Picozzi, A. Strollo, M. Pilz, D. Di Giacomo, B. Liss, and D. Bindi (2009). Are transients carrying useful information for estimating H/V spectral ratios? in Increasing Seismic Safety by Combining Engineering Technologies and Seismological Data, M. Mucciarelli (Editor), Proc. of the NATO Advanced Research Workshop, NATO Science for Peace and Security Series: C: Environmental Security, Springer, 17–31.
Rodriguez, V.H.S., and Midorikawa, S. (2002). Applicability of the H/V spectral ratio of microtremors in assessing site effects on seismic motion. Earthquake Engineering and Structural Dynamics, 31 (2), 261-279, DOI: 10.1002/eqe.108.
Author Response
Reviewer 1 comments (response below):
The authors develop a new method of processing microtremor data with the HVSRN method that uses the Hilbert Huang Transform, instead of the Fast Fourier Transform, to overcome problems of non-stationarity of the acquired signal. The applied methodology is described in a very rigorous way from a mathematical point of view and the approach used by the authors appears very convincing, both in terms of setting the method and in terms of the statistical analysis of the results. The choice not to use simulations, but only experimental data well constrained by geological knowledge, appears justified by the complexity of the physical problem which involves great difficulties for simulations that take into account too many physical variables.
In my opinion, therefore, the manuscript deserves to be accepted for publication on Sensors, after a minor revision that takes into account the points described below:
1) The introduction should also mention the debate among authors who consider it necessary to exclude spikes and transients in microtremors (e.g. Horike et al., 2001) and others (e.g. Mucciarelli and Gallipoli, 2003; Parolai et al., 2009) who suggest that nonstationary large amplitude noise windows should not be removed, because these can carry subsoil information, improving the correlation between noise and earthquakes in HVSRN curves. Furthermore, other authors suggest methods for the selection of time windows using statistical criteria that use agglomerative hierarchical clustering (e.g. Rodriguez and Midorikawa, 2002; D'Alessandro et al, 2016).
2) The authors should better clarify the reason for the excessive calculation time difference for the estimation of the HVSRN curve between the use of HHT compared to FFT (software not optimized or what?). In fact, this could greatly affect the preference of FFT over HHT, despite the undoubted advantages in terms of misfit obtained.
3) Even if, as mentioned above, the choice of testing the method with only experimental data appears quite justified, however the results of a single test site would not guarantee a robust evaluation of the effectiveness of the method. So I would suggest presenting at least one other test site, preferably in very different environmental conditions from the one described in the paper.
References of point 1:
D’Alessandro, A., D. Luzio, R. Martorana, and Capizzi, P. (2016). Selection of time windows in the Horizontal to Vertical Noise Spectral Ratio by means of cluster analysis. Bulletin of the Seismological Society of America, 106 (2), 560-574. doi: 10.1785/0120150017.
Horike, M., B. Zhao, and H. Kawase (2001). Comparison of site response characteristics inferred from microtremors and earthquake shear waves, Bull. Seismol. Soc. Am. 81, 1526–1536.
Mucciarelli, M., M. R. Gallipoli, and M. Arcieri (2003). The stability of the horizontal-to-vertical spectral ratio of triggered noise and earthquake recordings, Bull. Seismol. Soc. Am. 93, no. 3, 1407–1412.
Parolai, S., M. Picozzi, A. Strollo, M. Pilz, D. Di Giacomo, B. Liss, and D. Bindi (2009). Are transients carrying useful information for estimating H/V spectral ratios? in Increasing Seismic Safety by Combining Engineering Technologies and Seismological Data, M. Mucciarelli (Editor), Proc. of the NATO Advanced Research Workshop, NATO Science for Peace and Security Series: C: Environmental Security, Springer, 17–31.
Rodriguez, V.H.S., and Midorikawa, S. (2002). Applicability of the H/V spectral ratio of microtremors in assessing site effects on seismic motion. Earthquake Engineering and Structural Dynamics, 31 (2), 261-279, DOI: 10.1002/eqe.108.
=====
Response:
Dear Reviewer,
Thank you very much for the time and effort you spend in helping us to improve our manuscript. We hope that together we can create the best possible version of the manuscript for our readers.
Response to specific comments (relevant lines in the manuscript to each response are given in brackets behind our response):
1) We added your recommendation to our introduction. [p.2 ll.39-46]
2) Your intuition was correct, the software (partly just scripts) we use has not been optimised besides parallelisation. Much work could be done to reduce the computation time by clever reformulating the code. Our software is research code, not production code. We added a remark in the manuscript to clarify this. [p.7 ll. 245-247]
3) In order to compensate for the lack of synthetic data, we added an additional example. Even though the additional example remains inconclusive for the hypothesis presented in the manuscript, it does show two things: 1) it demonstrates that the result is not general (we mentioned and still mention that in the conclusions) but depends on the specific data, and 2) that very stringent conditions must be met by a data set in order to be a useful example for our verification strategy (i.e. using data inversion). Very accurate lithologic columns are required to corroborate inversion results conclusively and it is rare that HVSR data is available (to us) at the well log site. The ICJA site was a lucky exception, making it an ideal laboratory for our purpose. We hope that this has become more clear. [pp.9-13 ll.295-353]
Reviewer 2 Report
See the attached.

Author Response
Reviewer 2 comments (response below):
Thanks for your contributive work to Sensors. This paper proposes a method of using HHT instead of FFT to calculate HVSR with real data recorded at ICJA. MEMD is used instead of EMD to ensure that multivariate signal corresponds to a common spectral basis at all time. The result of HHT-based inversion yields a lower misfit. In general, the paper is good and well organized. Although the paper has already given a lot of detailed information of data processing, I am still confused at some points. I give some detailed comments in the following context which I hope can help the authors in the revision.
1. The authors think that “constructing realistic, complex synthetic ambient noise data is a daunting task ” ( Line 177), I think it is necessary to use theoretical simulation test to prove the superiority of the newly proposed method.
2. The distribution of noise sources in an urban environment is extremely complicated, especially in the case of short-time recording. The proposed HHT-based method has advantages over FFT-based method if the distribution of noise sources does not satisfy the condition of stationary phase, nevertheless, one day’s data is artificially selected for a permanently installed seismometer, so should the distribution and intensity of noise sources on this specifically selected day be analyzed?
3. No simulated test is presented, and only one day’s data and one station were used in the process of data processing. I don’t think such a case is of general significance. Is there a big gap in the result if it is replaced by data from another day or another station?
=====
Response:
Dear Reviewer,
Thank you very much for the time and effort you spend in helping us to improve our manuscript. We hope that together we can create the best possible version of the manuscript for our readers.
Response to specific comments (relevant lines in the manuscript to each response are given in brackets behind our response):
1) We expanded on our argumentation in the manuscript on why we do not use simulated data. In short, it is too complex to generate and would not generalise, too. In order to compensate for this, we added an additional example. Even though the additional example remains inconclusive for the hypothesis presented in the manuscript, it does show two things: 1) it demonstrates that the result is not general (we mentioned and still mention that in the conclusions) but depends on the specific data, and 2) that very stringent conditions must be met by a data set in order to be a useful example for our verification strategy (i.e. using data inversion). Very accurate lithologic columns are required to corroborate inversion results conclusively and it is rare that HVSR data is available (to us) at well log sites. The ICJA site was a lucky exception, making it an ideal laboratory for our purpose. We hope that this has become more clear. [pp.9-13 ll.295-353]
2) We still do not include any analysis of the signal and noise sources in the data we use. For the ICJA data example, in order to choose a quiet day, we statistically compared daily HVSR data for the entire year 2017, but we did not include that analysis here. Eastern Sunday was chosen, because it had one of the lowest daily variations in the HVSR curves. Still for the scope of the manuscript, data for a single day is more than what is usually used for HVSR data interpretation (2 to 4h is the norm) and choosing a quiet day is something a practitioner would do if he had the liberty to do so. The manuscript was written from the perspective of a potential practitioner and simply presents what would be the expected result for using HHT over (or better yet in conjunction with) FFT. Analysing the data further (i.e. for sources) would probably require us to increase the scope of the article beyond a single day's data and would add a significant amount of additional material to be covered. We feel that this would distract from the current focus. Moreover, there are no resources available to expand the scope of the manuscript. [no modifications]
3) We added an additional example from data of a very different site (see point 1). Data from another day would not change the outcome much, it would increase the HVSR curve confidence bounds and with it the obtained inversion model confidence. We selected the most quiet day of 2017 for the first example (see point 2) and a best guess for the second example (due to limited time for the preparation of this revision). For the first example, HVSR curves are very similar over the course of many days. [pp.9-13 ll.295-353]
Reviewer 3 Report
This paper proposes to use the Hilbert Huang Transform instead of FFT to obtain HVSR and demonstrates that with HHT the obtained HVSR is more robust.
Generally speaking, the paper is well organized and written.
The conclusion is supported by comparing 1D inversion results for FFT and HHT-based HVSR estimates from data measured at a well-studied, urban, permanent station. However, I am still confused. As the authors mentioned, it is difficult to compare two processing algorithms with real data, because the true result is generally not known. So why not use synthetic data? The authors illustrate that HHT can reduce biases, does it mean that HHT provides a smoother HVSR curve? If it does, will we lose some useful information on model parameters? Authors should know that a sharp peak and trough may appear in the HVSR curve and contains the information on model parameters (Mi et al., 2019) if there exists a high velocity contrast in the subsurface model.
If my confusion can be clarified, I think the paper can be accepted for publication.
Reference:
Mi, B., Y. Hu, J. Xia, and L. V. Socco, 2019, Estimation of horizontal-to-vertical spectral ratios (ellipticity) of Rayleigh waves from multistation active-seismic records: GEOPHYSICS, 84, EN81–EN92.
Author Response
Reviewer 3 comments (response below):
This paper proposes to use the Hilbert Huang Transform instead of FFT to obtain HVSR and demonstrates that with HHT the obtained HVSR is more robust.
Generally speaking, the paper is well organized and written.
The conclusion is supported by comparing 1D inversion results for FFT and HHT-based HVSR estimates from data measured at a well-studied, urban, permanent station. However, I am still confused. As the authors mentioned, it is difficult to compare two processing algorithms with real data, because the true result is generally not known.
1) So why not use synthetic data?
2) The authors illustrate that HHT can reduce biases, does it mean that HHT provides a smoother HVSR curve? If it does, will we lose some useful information on model parameters? Authors should know that a sharp peak and trough may appear in the HVSR curve and contains the information on model parameters (Mi et al., 2019) if there exists a high velocity contrast in the subsurface model.
If my confusion can be clarified, I think the paper can be accepted for publication.
Reference:
Mi, B., Y. Hu, J. Xia, and L. V. Socco, 2019, Estimation of horizontal-to-vertical spectral ratios (ellipticity) of Rayleigh waves from multistation active-seismic records: GEOPHYSICS, 84, EN81–EN92.
=====
Response:
Dear Reviewer,
Thank you very much for the time and effort you spend in helping us to improve our manuscript. We hope that together we can create the best possible version of the manuscript for our readers.
Response to specific comments (relevant lines in the manuscript to each response are given in brackets behind our response):
1) We expanded on our argumentation in the manuscript on why we do not use simulated data. In short, it is too complex to generate and would not generalise, too. In order to compensate for this, we added an additional example. Even though the additional example remains inconclusive for the hypothesis presented in the manuscript, it does show two things: 1) it demonstrates that the result is not general (we mentioned and still mention that in the conclusions) but depends on the specific data, and 2) that very stringent conditions must be met by a data set in order to be a useful example for our verification strategy (i.e. using data inversion). Very accurate lithologic columns are required to corroborate inversion results conclusively and it is rare that HVSR data is available (to us) at the well log site. The ICJA site was a lucky exception, making it an ideal laboratory for our purpose. We hope that this has become more clear. [pp.9-13 ll.295-353]
2) In our experience HHT results are more smooth than raw FFT results. However, smoothness and peak amplitudes depend very much on processing parameters for FFT and are much more stable for HHT results. Troughs are usually more similar (please compare with the examples) between HHT and FFT. That said, relatively sharp peaks with larger amplitudes than what are presented here are possible for HHT results, we simply do not have well log information for those examples with which to corroborate the inversion results (and therefore do not include them). As you say it is true, that peak amplitude contains valuable information about model parameters. However, we would argue that, especially, peak amplitude is very poorly resolved data with FFT and depends massively on the processing parameters (for which does not exist one best set) window length, window type, smoothing type and smoothing length. This limitation (of having to choose these parameters) does not apply to HHT. Furthermore, we demonstrate that it is easier to fit a model to HHT results than to FFT results, the latter often estimate especially the confidence limits poorly. We hope that we clarify your doubts with the additional example that we included in the revised version. [pp.9-13 ll.295-353]